# Potential of the Use of Biostimulants in Lettuce Production

**DOI:** 10.3390/plants14152416

**Published:** 2025-08-04

**Authors:** Talys Moratti Lemos de Oliveira, Janyne Soares Braga Pires, Vinicius de Souza Oliveira, Ana Júlia Câmara Jeveaux Machado, Adriano Alves Fernandes, Lúcio de Oliveira Arantes, Sara Dousseau-Arantes

**Affiliations:** 1Federal Institute of Espírito Santo (IFES), Barra de São Francisco Campus, Rodovia ES 320, Km. 118, Zona Rural, Barra de São Francisco 29800-000, ES, Brazil; talys.oliveira@ifes.edu.br; 2Federal University of Espírito Santo (UFES), Center for Natural Human Sciences (CCHN), Goiabeiras Campus, Vitória/ES. Av. Fernando Ferrari, 514, Vitória 29075-910, ES, Brazil; janynesbraga2@gmail.com (J.S.B.P.); anajucamara@gmail.com (A.J.C.J.M.); 3Capixaba Institute of Research, Technical Assistance and Rural Extension, Rodovia BR-101 Norte, Km. 151, Bebedouro, Linhares 29915-140, ES, Brazil; souzaoliveiravini@gmail.com (V.d.S.O.); lucio.arantes@incaper.es.gov.br (L.d.O.A.); 4Federal University of Espírito Santo (UFES), Centro Universitário Norte do Espírito Santo (CEUNES), São Mateus Campus, Rodovia BR 101-Norte, Km. 60, Litorâneo, São Mateus 29932-540, ES, Brazil; adriano.fernandes@ufes.br

**Keywords:** *Lactuca sativa* L., humic substances, sustainability

## Abstract

Lettuce (*Lactuca sativa* L.) is one of the main leafy vegetables in the world, being present in several countries. Due to its composition, which includes a substance with antioxidant action and beneficial effects on health, it is consumed constantly. However, due to ongoing climate change that has had global effects, the crop has been suffering a reduction in productivity and quality. Thus, technologies aiming to mitigate the effects of climate extremes have been developed. In lettuce production, biostimulants make it possible to improve the growth and sustainable development of plants. This is due to their ability to activate physiological and biochemical processes in plants, resulting in an increase in the production of bioactive compounds such as vitamins, amino acids, and antioxidants. In addition, biostimulants contribute to improving the nutritional quality of lettuces, making them more resistant and adapted to different environmental conditions, resulting in a more sustainable development for the crop. This review aims to compile and discuss the available scientific evidence on the use of biostimulants in lettuce cultivation, addressing their mechanisms of action, the types of substances used, the results obtained in different cultivation systems, and their potential to promote more efficient and adaptable agriculture in the face of environmental changes.

## 1. Introduction

Lettuce (*Lactuca sativa* L.) is a leafy vegetable belonging to the Asteraceae family; it stands out as one of the crops with the greatest economic and nutritional importance worldwide [1]. Generally, lettuce varieties are grown for leaf production, so that the life cycle is interrupted before flower formation, preventing seed maturation and ensuring the quality of the leaves for consumption [2].

Lettuce is widely known and consumed across many regions of the world [3]. However, the growing of lettuce faces challenges related to climate change, such as heat waves, high temperatures, and nutrient solution salinity, which impact productivity and product quality [3,4,5,6,7]. In view of this scenario, the use of a substance with biostimulant effects appears as a promising avenue to mitigate the effects of abiotic stress on lettuce production [5,6,7].

Biostimulants are available in various forms, including plant extracts, seaweed extract, protein hydrolysates, vitamins, amino acids, non-microbial compounds, humic acids, fulvic acids, and their derivatives [7,8,9,10,11,12,13,14,15,16]. These substances have already shown a beneficial effect on several plant species, boosting growth and development, improving the absorption and use of nutrients, minimizing the effects of biotic and abiotic stress, improving hormone synthesis, and minimizing the effects of free radicals [7,8,9,10,11,16].

Considering these aspects, conducting a review on the use of biostimulants in lettuce cultivation becomes essential to consolidate current knowledge, identify research gaps, and guide future studies aiming to improve crop productivity and sustainability. Thus, the present review aims to deepen the knowledge on lettuce cultivation by addressing the influence of biostimulants on agronomy, aiming to contribute to the development of more sustainable and efficient agricultural vegetables.

## 2. Biostimulants in Lettuce Cultivation

The lettuce crop has been facing constant challenges, especially in relation to climate extremes [3]. Climate change has caused significant impacts on lettuce production, raising costs, reducing productivity, and limiting supply in the consumer market [3]. Extreme weather events, such as torrential rains and prolonged droughts, have forced farmers to seek alternative solutions, such as adopting protected crops [7]. In this context, the use of technologies that allow the mitigation of the effects of climate on crops through practical measures and allow more sustainable crops becomes necessary to ensure food security where the weather conditions are unfavorable [6,7,8,9,10].

Several compounds are tested in order to generate biostimulant effects in plants and optimize their growth and development [7,10]. These compounds act by stimulating plant metabolism, increasing enzyme activity and protein synthesis. They trigger physiological defense responses, such as increased antioxidant production and the activation of protection systems against environmental stresses [7,8,9,10]. Furthermore, they help in hormonal regulation, favoring cell division and tissue stretching, which contributes to a more balanced and efficient development. In addition, biostimulants can play a crucial role in nutrient uptake and translocation, optimizing the utilization of available resources and maximizing the productive potential of crops [7,9,10,11].

In light of these considerations, the integration of biostimulant technologies into lettuce cultivation emerges not only as an innovative agronomic tool, but also as a strategic response to the increasing vulnerability of crops under climatic pressure. By reinforcing physiological resilience and enhancing nutrient efficiency, biostimulants contribute to stabilizing yields and maintaining crop quality, even under suboptimal environmental conditions (Figure 1). As climate variability continues to intensify, adopting such adaptive measures becomes essential—not merely to improve production outcomes but to sustain agricultural systems in a future marked by uncertainty and a growing demand for resilient, high-quality food.

### 2.1. Humic Substances

Humic substances have in their composition organic material resulting from the decomposition of plant and animal remains that are used in the formulation of fertilizers for various crops [10,12,13,15]. They use is related to positive effects both on the characteristics of the substrate and on the plant’s metabolism. Because they are made up of humic acids, fulvic acids, and humins, humic substances stimulate the production of hormones important for plants, such as auxin, cytokinins, and gibberellins, which can influence metabolic processes favorably, favoring plant growth and development [5,11,12,14,17,18,19,20]. Humic substances act as amphiphilic redox compounds with outstanding chelating properties, contributing to soil health by increasing nutrient bioavailability, stabilizing soil aggregates, and enhancing plant growth by promoting gains in chlorophyll levels, the synthesis of nucleic acids, and improvement in water and nutrient absorption [10,13,14,17].

The application of humic acids, plant growth-promoting bacteria, and their combination positively influenced several agronomic traits of lettuce cv. Vanda [13]. The isolated application of bacteria increased plant height by 18.85%, while the combined treatment yielded the best results for the length of the largest leaf, rosette circumference, and shoot fresh weight, with an increase of up to 33.5% compared to the control [13]. The application of a combined biostimulant containing plant-growth-promoting bacteria (PGPB) and *Chlorella vulgaris* significantly improved the yield and quality of lettuce [14]. When applied every 14 days, this treatment increased the fresh weight of romaine lettuce by 18.9% in the spring crop and leaf lettuce by 22.7% in the summer crop [14]. These biostimulants are low-cost, sustainable technologies with high potential to improve lettuce productivity under field conditions [20].

Taken together, the evidence reinforces the functional versatility of humic substances and microbial biostimulants in enhancing lettuce cultivation [12,14,15]. Their synergistic action—whether through improved nutrient dynamics, hormonal stimulation, or structural and biochemical modifications in the plant—offers a practical and ecologically sound alternative to conventional inputs. By promoting key growth parameters and increasing yield components, these treatments not only support higher productivity but also contribute to the sustainability of horticultural systems. As such, humic-based and microbial biostimulants stand out as effective tools for meeting the challenges of modern agriculture, particularly in the face of resource limitations and the growing need for environmentally responsible practices.

### 2.2. Hydrolysates and Nitrogen Compounds

Another group of biostimulant substances used in agriculture are protein hydrolysates and nitrogen-containing compounds. This group of amino acids has been widely used, and its functions in plants are associated with increased nitrogen assimilation, hormone synthesis, antioxidant effects, increased nutrition, and biomass accumulation in plants [8,9,10,16]. These substances have a central carbon in their composition, which is mostly attached to a carboxyl group (COOH), an amino group (NH_2_), and a hydrogen atom. Amino acids in plants are also related to protein synthesis, are hormone precursors, and promote tolerance to water, heat, and salinity stress and pest and disease attack [8,9,17].

In recent years, studies using protein hydrolysates and nitrogen-containing compounds have proven their effectiveness as biostimulant substances. Protein hydrolysates (PHs) improve antioxidant enzyme activity and promote the synthesis of secondary metabolites, which contribute both to plant stress tolerance and to the nutraceutical qualities of lettamiuce [9,16,18]. The application of seaweed extract (SWE) and protein hydrolysates (PHs) as biostimulants in lettuce grown under saline (40 mM NaCl) and non-saline (0 mM NaCl) conditions [21]. Both biostimulants improved plant growth and shoot fresh weight, even under salt stress. Protein hydrolysates (PHs) and seaweed extract (SWE) reduced the sodium content under salinity by 15.6% and 9.4%, respectively, while PH also significantly lowered chloride levels [21]. Additionally, the treatments triggered broad metabolic reprogramming related to stress tolerance, promoting the accumulation of glucosinolates, phytoalexins, and jasmonates.

The aplication of commercial plant biostimulants based on plant-derived protein hydrolysates—Viva^®^, Radifarm^®^, and Megafol^®^—on the growth of red and green romaine lettuce grown under controlled conditions has been reported in the literature [17]. Although biostimulant application did not significantly affect the final plant height, it positively influenced other growth parameters such as leaf number, leaf area, shoot fresh weight, and chlorophyll content. Radifarm^®^ treatment resulted in the highest shoot and root fresh weights, while Viva^®^ showed the best fit in the polynomial growth model for plant height [17]. The green variety exhibited a higher total chlorophyll content and greater leaf area compared to the red variety. These findings highlight the potential of biostimulants composed of amino acids, vitamins, polysaccharides, and humic acids to enhance physiological processes related to photosynthesis and biomass accumulation, contributing to more efficient and sustainable production systems.

Specifically for lettuce crops, the application of amino acids has shown positive effects on plant growth and development, in addition to significantly contributing to processes related to photosynthesis and improved nutrient amounts [7,8,10]. The exogenous application of amino acids, particularly glycine, methionine, and proline, has been demonstrated to modulate key physiological processes in lettuce plants, such as enhancing pigment biosynthesis, improving photosynthetic efficiency, and optimizing ion homeostasis. These metabolic adjustments not only contribute to the enhancement in biomass production and nutritional quality but also play a crucial role in alleviating the detrimental effects of salt stress that are commonly encountered in lettuce cultivation [22,23].

Moreover, amino-acid-based biostimulants play a multifaceted role beyond their direct nutritional contributions. These molecules function as the precursors of phytohormones, osmolytes, and signaling compounds, modulating plant responses at the cellular and molecular levels. Their exogenous application has been associated with improved nitrogen metabolism, enhanced chlorophyll biosynthesis, and the stimulation of antioxidant enzymes, contributing to greater photosynthetic efficiency and biomass accumulation [5,6,7,8,9,22].

Studies using enzyme-based biostimulants have reported increases in fresh and dry biomass, SPAD index, and the accumulation of health-promoting compounds in lettuce leaves [23]. Moreover, plant-based biostimulants are also being studied in lettuce cultivation as biostimulants. Studies using 6% moringa leaf extract, showed positive results in lettuce head size and weight, as well as a higher number of leaves—interesting traits for this species, which is commercially valued for attributes such as head size [24].

These substances have been proven effective in enhancing photosynthetic activity, stimulating antioxidant responses, improving ion balance under stress, and ultimately promoting greater biomass accumulation and crop quality. Among the most relevant observations are the consistent increases in leaf area, shoot fresh weight, chlorophyll content, and stress resilience, particularly under adverse conditions such as salinity or high temperature. Additionally, the exogenous supply of amino acids has been associated with improved nitrogen metabolism and hormonal modulation, contributing not only to productivity but also to the physiological stability of plants. The evidence reinforces that these compounds do not act merely as nutrient supplements but operate through complex regulatory mechanisms, supporting more efficient and sustainable lettuce production systems. Given the commercial importance of traits such as head size, leaf number, and biomass in lettuce, the strategic integration of biostimulants into cultivation practices presents itself as a valuable tool for optimizing yield while maintaining quality under both controlled and field conditions.

### 2.3. Algal Extracts and Chitosan

Among the compounds that perform a biostimulatory function, algal extracts act on the oxidative system of plants, achieving a greater tolerance to adverse environmental conditions and inducing a better regenerative capacity after being exposed to these conditions, which favors productive gains or is maintained even under unfavorable conditions [5,14,17,20,25]. These compounds are an important alternative for crops with greater sustainability, since they do not have a negative impact on the environment. In addition, algal extracts have reduced costs, are easy to prepare and handle, are efficient in small dosages, and contain in their composition macro- and micronutrients, amino acids, vitamins, carbohydrates, and plant growth hormones [5,14,25,26].

The algal extract from *Ascophyllum nodosum* was tested in hydroponic lettuce cultivation [26,27]. The authors found that the application of *A. nodosum* alters the negative effects of potassium deficiency during plant growth and development. In addition, the use of *A. nodosum* positively altered the activity of protective enzymes such as superoxide dismutase, catalase, and peroxidase, which act against the formation of free radicals such as hydrogen peroxide (H_2_O_2_). It was also possible to observe that the application of *A. nodosum* promoted an increase in quality and a decrease in the respiratory rate of the plants, thus increasing the shelf life of lettuce [26,27]. Additionally, the combined application of *A. nodosum* and humic acid has been shown to promote initial growth and mitigate post-harvest losses, probably because of increased biomass and antioxidant capacity, which are essential factors for quality preservation during storage [25,26,27,28]. *Lithothamnium* sp. is another seaweed whose application in agriculture is successfully reported. It has calcium carbonate, humic and fulvic substances, and more than 20 compounds in its composition [29,30]. The study demonstrated its biostimulant effectiveness in agricultural crops [30].

Chitosan has demonstrated promising effects as a plant biostimulant in lettuce, acting at multiple physiological and biochemical levels [31,32,33,34,35]. The foliar application of chitosan at varying concentrations (0, 75, 150, and 300 ppm) has been shown to significantly improve yield, enhance chlorophyll content, and improve leaf quality [31]. Notably, a 300 ppm concentration resulted in a marked reduction in nitrate accumulation, one of the primary metabolic contaminants in leafy vegetables, as reported in recent studies [31].

Beyond its isolated use, chitosan has also been employed in association with iodine-based compounds such as potassium iodate (KIO_3_), forming complexes like chitosan-KIO_3_ [32]. This combination not only promotes increased biomass accumulation but also enables iodine biofortification without negatively impacting crop performance. Experimental findings indicate that such complexes maintain lettuce productivity while elevating the iodine concentrations in plant tissues, thus contributing to human nutritional security [32].

During the early stages of development, the application of chitosan hydrolysate for seed treatment has proven to be effective in promoting germination and stimulating root growth [33,34]. Recent data show increases in fresh biomass, chlorophyll, and carotenoid contents, along with enhanced root branching, critical traits for successful crop establishment [32,33,34]. These effects highlight the role of chitosan as a modulator of primary metabolism and a promoter of photosynthetic efficiency [31,32,33,34,35].

Under saline stress conditions, chitosan continues to function effectively as a biostimulant [34]. Its foliar application supports the maintenance of leaf area and biomass in both shoot and root systems while enhancing physiological stability through the activation of antioxidant enzymes and regulation of ion balance and osmotic adjustment [34]. These findings suggest that chitosan plays a role in the adaptive mechanisms associated with abiotic stress, potentially through hormonal signaling and the induction of secondary metabolic pathways.

Moreover, the combination of plant-growth-promoting bacteria and microalgae can act positively under heat stress, enhancing plant productivity and antioxidant capacity even under high-temperature conditions [14,20,24]. Such combinations have been effective in improving shoot biomass and total carotenoid content, highlighting their potential for sustainable lettuce production during periods of climatic stress [20].

More recently, the use of iodine-associated nanochitosan complexes (NPSCS-i) has shown superior responses compared to conventional formulations [35]. Under controlled conditions, the application of 5 mg L^−1^ of KIO_3_ in a nanostructured formulation resulted in a 30.8% increase in lettuce yield, underscoring the potential of combining nanotechnology with natural polymers to sustainably intensify agricultural production [35].

In this context, the application of chitosan has emerges as a promising strategy to optimize the physiological performance of lettuce and enhance its productive efficiency, particularly in agricultural systems aiming for greater sustainability. The wide range of observed effects—from the improved germination to increased biomass and enhanced nutritional quality of leaves—demonstrates that chitosan does not act in isolation [31,32,33,34,35], but rather interacts with multiple physiological and metabolic pathways, enabling the plant to adapt to various environmental conditions.

Its effectiveness under saline stress, for instance, highlights a regulatory role that extends beyond basic nutrition and involving antioxidant defense mechanisms and osmotic balance [34]. Additionally, the formation of iodine-based complexes, especially in nanostructured formulations, represents a significant advancement in both biofortification and the maintenance of crop productivity. This indicates that chitosan, beyond serving as a valuable tool in agronomic management, may also contribute to strategies focused on improving the nutritional and functional quality of food.

The consistency of results across different modes of application—whether via seed treatment, foliar spraying, or in combination with micronutrients—further reinforces its potential as a versatile and efficient biostimulant. Therefore, its use in lettuce cultivation not only supports immediate agronomic gains but also aligns with ecologically based agriculture, reducing the dependence on synthetic inputs and promoting the adoption of clean, integrated technologies.

### 2.4. Inorganic Compounds

Among the main inorganic compounds that play a beneficial role in plant development are Al, Co, Na, Se, and Si [36,37]. These elements help promote growth but are not essential for plants [37]. The positive effects on plants may be related to silica deposition in the cell wall under stressful conditions such as pathogen attack or high salinity [36,37]. Studies prove that silicon plays a role in mitigating biotic and abiotic stress in plants grown in hydroponic systems [37]. In studies carried out with *Lithothamnium* sp. algal extract, it is reported that a possible cause of the improvement in the quality of lettuce production is related to carbonate-siliciclastic sea sand being incorporated during the extraction of the algae, which contains silicon [29,30].

These findings reinforce the relevance of certain inorganic elements—particularly silicon—in supporting plant development, even though they are not classified as essential nutrients [36,37]. In lettuce cultivation, especially under hydroponic conditions, silicon has shown consistent benefits by enhancing nutrient uptake, strengthening structural defenses, and promoting tolerance to stress factors. The presence of silicon in natural sources such as *Lithothamnium* sp. may further explain improvements in crop quality when these inputs are used [29,30]. Overall, incorporating silicon-based compounds into lettuce production systems appears to be a promising strategy to boost both growth and physiological resilience.

### 2.5. Fungi

In the case of biostimulants that include fungi in their composition, *Trichoderma*-based products are the most used in Brazil and Latin America. The practical application of *Trichoderma* lies in the fungus’s ability to manage disease, compete with pathogenic agents for energy sources, and produce antibiotics that inhibit the proliferation of the pathogen [38]. Another important function of *Trichoderma* is to stimulate the production of plant hormones, such as jasmonic acid, salicylic acid, and ethylene, which induce plant resistance [19,20,39,40,41].

Successful cases using *Trichoderma* as a biostimulant were reported with a 23.7% increase in the production of commercial pepper fruits (*Capsicum annuum* L.) [19]. It was also possible to observe positive effects on vegetative and reproductive growth; the synthesis of hormones such as gibberellins, auxin, and cytokinin; and the induction of the formation of secondary compounds such as carotenoids, saponin, and phenolic compounds [19]. In carrot crops, the application of *Trichoderma* to the seeds promoted greater plant growth, in addition to presenting similar effectiveness to treatment with agrochemicals in terms of reducing the incidence of pathogenic fungi such as *Alternaria dauci*, *Alternaria radicina*, *Rhizoctonia solani*, and *Sclerotinia sclerotiorum* [19]. For lettuce crops, the addition of *Trichoderma* provided an improvement in the nutritional status of the leaf, with a greater accumulation of potassium and magnesium, in addition to a higher yield of fresh and dry biomass. It was also found that the application of *Trichoderma* provided greater antioxidant activity, increased total ascorbic acid and total phenols, and lowered the nitrate content [19,40].

In a study with lettuce plants, a biostimulan derived from the solid-state fermentation of green residues and wood chips with *Trichoderma harzianum* was evaluated, with and without L-tryptophan supplementation [40]. The product improved lettuce growth parameters such as fresh weight, plant height, leaf area, and antioxidant compounds, especially under proper irrigation [40].

Altogether, the promising results observed with *Trichoderma*-based formulations, as well as their combinations with other microbial agents, reinforce their value as multifunctional biostimulants capable of enhancing both productivity and crop quality [19,20,38,39,40,41]. Their ability to induce resistance mechanisms, promote hormonal balance, and stimulate the accumulation of secondary metabolites places these fungi-based products among the most versatile tools available in sustainable horticulture. In lettuce cultivation specifically, their effects go beyond simple yield increases, contributing meaningfully to the nutritional and functional quality of the harvested product. As agricultural systems face increasing environmental and economic pressures, expanding the understanding and use of biostimulants such as *Trichoderma* has become a necessary step toward resilient and efficient production models.

## 3. Perspectives

Continued advancements in the use of biostimulants for lettuce cultivation opens up promising avenues for more sustainable and resilient agricultural systems. Although considerable progress has been made in identifying compounds that improve growth, quality, and stress tolerance in lettuce, the complexity of interactions between bioactive molecules, environmental factors, and plant physiology still demands more in-depth investigation. The current body of research—summarized in Table 1—demonstrates the diversity of the available biostimulant classes, from humic substances and protein hydrolysates to seaweed extracts, microbial consortia, and enzyme-based formulations, each offering distinct physiological benefits to the crop.

Although the beneficial effects of biostimulants in mitigating abiotic stress and enhancing plant growth are well recognized, their effective adoption in lettuce cultivation requires a careful and context-specific approach. The physiological mechanisms activated by biostimulants vary widely depending on the type of compound used—such as humic substances, seaweed extracts, amino acids, protein hydrolysates, or growth-promoting microorganisms—as well as factors like application rate, timing, and environmental conditions. This complexity often limits the consistency of agronomic outcomes, especially when transitioning from controlled experimental conditions to the variability of open-field environments. Moving forward, integrating biostimulants into commercial production systems will require a more precise understanding of dose–response relationships, application timing, and synergistic effects between compounds. Furthermore, as climate change continues to intensify abiotic stresses such as salinity, heat, and water scarcity, the role of biostimulants in mitigating these effects is becoming increasingly relevant.

In this context, despite aligning with sustainable agricultural practices and the goal of reducing chemical input dependence, the technical and economic feasibility of biostimulants in lettuce production depends on further research. Large-scale field validation is necessary to confirm the promising effects observed in preliminary studies. A critical perspective suggests that the widespread integration of biostimulants will require investment in technological innovation to improve formulations, optimize delivery systems, and deepen understanding of the interactions between bioactive compounds, abiotic stressors, and plant physiology. Beyond yield enhancement, biostimulants also hold the potential to improve nutritional quality, increase stress tolerance, and promote adaptive responses, positioning them as valuable tools for modern horticulture—though still in need of consistent and reproducible scientific validation across diverse agricultural systems.

Investment in research and innovation is therefore essential, not only to validate emerging products under diverse environmental and cultivation conditions but also to refine formulations and delivery technologies. The demand for more functional, nutrient-rich, and environmentally sustainable vegetables also adds pressure for solutions that improve both productivity and food quality. In this sense, the use of biostimulants aligns with global trends toward clean agriculture, a reduced reliance on synthetic inputs, and an enhanced nutritional efficiency.

## 4. Conclusions

In lettuce cultivation, biostimulants have stood out as a promising strategy, promoting more sustainable growth, higher productivity, and greater plant resistance to abiotic stresses. The use of these substances improves nutrient absorption and optimizes physiological processes such as photosynthesis and the synthesis of bioactive compounds. However, there are still gaps to be filled, such as the lack of specific regulation for the use of biostimulants and the need for more in-depth studies on their effects. Therefore, it is essential to invest in research that investigates the efficiency of different types of biostimulants, their mechanisms of action, and the economic and environmental impacts of their application in lettuce. This will enable the development of more accessible and regulated technologies, ensuring that the benefits of biostimulants are known and exploited.

## Figures and Tables

**Figure 1 plants-14-02416-f001:**
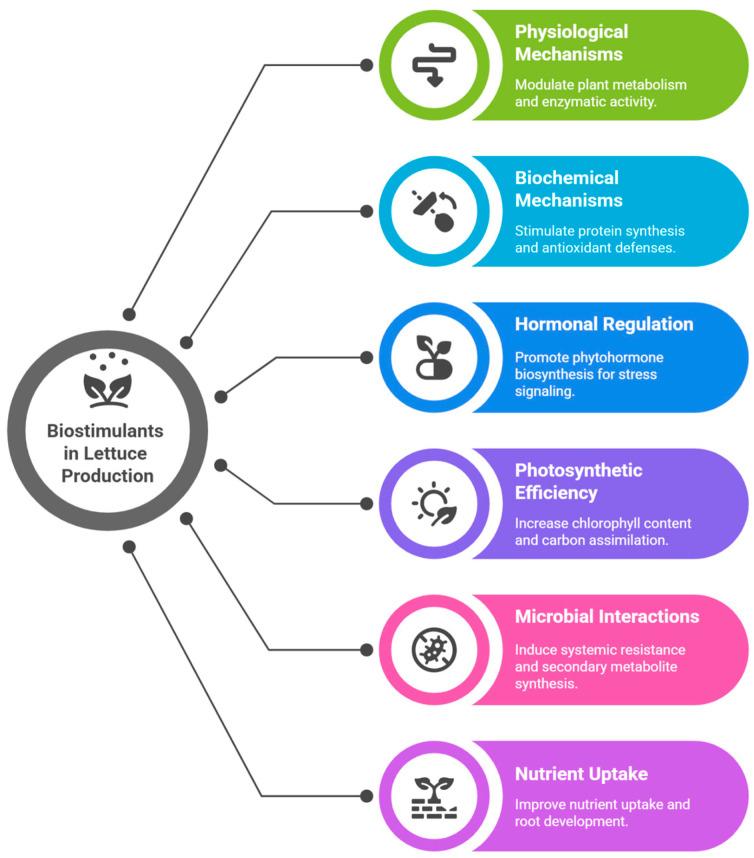
Biostimulants in lettuce production.

**Table 1 plants-14-02416-t001:** Comparative analysis of biostimulant applications in lettuce based on published scientific literature.

Class	Product	Authors	Effects on Lettuce
Humic and Fulvic Acids	BLACKJAK^®^	[12]	Increased biomass, antioxidant.
	Humic and Fulvic acids	[18,20,28]	Nutrient absorption, stress resistance.
Protein Hydrolysates	Trainer^®,^, Vegamin^®^ Viva^®^, Radifarm^®^, Megafol^®^ PHs	[16,17] [16,39]	Leaf area, number of leaves, shoot fresh weight (SFW), and chlorophyll content.
			Antioxidant activity and secondary metabolites. Shoot fresh weight, dry weight, leaf area, and yield.
Algal Extracts	*Ascophyllum nodosum* *Sargassum* spp. *Lithothamnium* sp.	[26,27,28] [27] [30]	Increase in quality, shelf life, protective enzymes, and biomass under salt stress.
			Accumulated dry biomass.
			Root development.
Bacteria + Microalgae	PGPR + *Chlorella vulgaris*	[13,14]	Shoot biomass and carotenoid content under heat stress. Lettuce yield.
Enzymes	Β-glucosidase + xylanase	[23]	Increase in fresh and dry biomass and bioactive compounds.
Plant Extracts	Moringa Leaf Extract (MLE)	[26]	Increase in head size and number of leaves.
Fungi-based	*Trichoderma* spp.	[19,38,40,41]	Increase in biomass, antioxidants, ascorbic acid, morphological and biomass parameters.

Source: Author’s own elaboration based on published studies, 2025.

## Data Availability

The original contributions presented in the study are included in the article; further inquiries can be directed to the corresponding authors.

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
