# Peer review of "Potential of the Use of Biostimulants in Lettuce Production"

_plants, 2025, doi:10.3390/plants14152416_

Round 1
Reviewer 1 Report (Previous Reviewer 2)
Comments and Suggestions for Authors
Biostimulants make it possible to improve the growth and sustainable development of plants. This is due to its ability to activate physiological and biochemical processes in plants, which resulting in an increase in the production of bioactive compounds such as vitamins, amino acids, and antioxidants. In this review, biostimulants contribute to improving the nutritional quality of lettuces, making them more resistant to pests and diseases. Plants that are more adapted to different environmental conditions result in a more sustainable development of the crop. the paper can be accepted after revised. the revised suggestion is as followed:
1.paper can be methodization after revised;
2. 2.Lactuca sativa L. may change into Lettuce vegetable;
3.part 3 can be divided into more subpart, so that it is easy to be clear understand;
4.the subpart can be 128 line 3.1 Humic substances; 168 line 3.2 protein hydrolysates and nitrogen-containing compounds; 230 line 3.3 seaweed extracts and chitosan ; 276 line 3.4 inorganic compounds; 291 line 3.5 fungi and their composition.
5. the 6.Conclusion can be changed into 4.Conculusion.
Author Response
Dear Reviewer,
Thank you very much for your insightful comments and constructive suggestions, which greatly contributed to improving the structure and clarity of our manuscript.
Please note the following in response to your observations:
1. Methodological organization: We restructured the article to improve its methodological flow. The section “3. Biostimulants in Lettuce Cultivation” has been reorganized into clearly defined subsections, as you recommended.
2. Lactuca sativa L.: We adjusted the text to use “lettuce” in place of Lactuca sativa L. in appropriate contexts to ensure better readability while maintaining scientific accuracy.
3. Section division (3.1 to 3.5): As suggested, we introduced new subheadings to structure Section 3, namely
- 3.1 Humic substances;
- 3.2 Hydrolysates and nitrogen compounds;
- 3.3 Algae extracts and chitosan;
- 3.4 Inorganic compounds;
- 3.5 Fungi.
This division facilitates reader understanding and improves the logical organization of the content.
4. Conclusion numbering: The final section was renumbered to Section 4 to maintain coherence with the previous edits.
5. Additional improvements: We also made other important adjustments, including
- The incorporation of more recent references (2020–2024);
- The removal of repetitive or general content regarding lettuce productivity and cultivation;
- The addition of a more critical and reflective perspective at the end of the Discussion;
- The formatting of references according to the guidelines for the journal Plants.
Once again, we appreciate your contribution to improving the quality and scientific rigor of our manuscript.
Sincerely,
The Authors
Reviewer 2 Report (Previous Reviewer 3)
Comments and Suggestions for Authors
Dear authors i have reviewd this review article and some major changes need to be made.
First english quality is poor, as example in line 52-54: Thus, the present review aims to deepen the knowledge about lettuce cultivation by addressing the influence of biostimulants on agronomic, aiming to contribute to the development of more sustainable and efficient agricultural vegetables. Agronomic what?
The abstract also has poor english, and it does not present what is the excat point of this rewiev article.
Line 109-119: Lack of citations, in general the citations are 10 years old. I storngly suggest citing literature form last 3 to 4 years.
line 244: H2O2 should be H2O2
Line 320,321: italic for latin names.
Table 1: Authors incorporated only 15 studuies, which i suggest to expend. There are only half of them above the year 2020 (which was already 5 years ago).
Comments on the Quality of English Language
Revise all english.
Author Response
Dear Reviewer,
We sincerely appreciate your thorough and constructive feedback. Your insightful comments were extremely helpful in improving the scientific quality, clarity, and structure of our manuscript.
In response to your specific suggestions, we made the following revisions:
1. English language editing: We requested a full professional review of the manuscript through the Rapid English Editing service on the MDPI Author Services (SuSy) platform, ensuring improved grammar, style, and clarity throughout the text.
2. Abstract revision: The Abstract was completely reformulated to clearly and concisely reflect the purpose of the review, the main focus of the content, and the overall contribution of the article.
3. Language improvement in key sections: We revised problematic sentences such as those in lines 52–54, aiming for greater precision and scientific clarity. Ambiguous terms were corrected, and redundant phrases were removed.
4. Inclusion of more recent references: We updated the literature base by incorporating studies primarily published in the last 3 to 4 years (2020–2024). This applies notably to Section 3 and Table 1, with emphasis on current research on biostimulants in lettuce cultivation.
5. Addition of subheadings within Section 3: Following your recommendation, the main section on biostimulants was divided into clear and topic-specific subsections (3.1 to 3.5). This restructuring enhances logical flow and readability.
6. Reduction in redundant content: We removed repetitive statements about lettuce productivity, distribution, and cultivation challenges that appeared in various sections of the manuscript. The focus was refined to better align with the article's objective.
7. Increased critical analysis: We expanded the final section to include a more critical and reflective discussion of the literature, limitations, and research gaps. The manuscript now emphasizes not just descriptive content but also synthesis and interpretation.
8. Chemical formatting and Latin names: Corrections were made to properly format chemical symbols (e.g., Hâ‚‚Oâ‚‚) and italicize Latin species names throughout the text.
9. Formatting references in journal style: All references were revised to comply with the Plants journal formatting requirements.
We sincerely thank you for helping us improve the rigor and overall quality of this review. We hope that the revised manuscript now meets your expectations and is better aligned with the journal's standards.
Reviewer 3 Report (Previous Reviewer 4)
Comments and Suggestions for Authors
The responses to reviewers and the submitted revised manuscript do not correspond. For instance, I found no modifications in the abstract that match the responses. Section 3 ("Biostimulants in Lettuce Cultivation," spanning 5 pages) still lacks any subheadings. Similar issues exist with other items. Please verify whether an incorrect version was submitted.
Given that the resubmitted manuscript inadequately addresses the previous review comments, the following suggestions pertain solely to the current version:
1. The title "2. Lactuca sativa L." is inappropriate. Moreover, this section contains excessive descriptions of plant characteristics and industrial development while lacking substantive content relevant to the paper's focus, requiring further refinement.
2. The manuscript suffers from significant content repetition, particularly regarding lettuce's importance, yield, distribution, and cultivation challenges. These redundancies appear throughout the text and should be eliminated.
3. Substantially expand Table 1 while appropriately condensing related textual descriptions to improve conciseness. Focus textual analysis on summarizing and evaluating research progress to enhance the manuscript's depth. The current version remains akin to a simple literature catalog, lacking the comprehensive synthesis and analytical depth expected in a review article.
4. Avoid duplicating content between tables and text.
5. Propose generalized modes of action or include schematic diagrams summarizing the comprehensive mechanisms of biostimulants.
6. If this represents the revised version, corresponding improvements based on the initial review comments remain necessary.
Author Response
Dear Reviewer,
Thank you very much for your careful and critical assessment of our manuscript. Your detailed comments were fundamental for enhancing the coherence, structure, and analytical depth of our article.
Here are the main adjustments made in response to your valuable suggestions:
1. Resubmission verification: We sincerely apologize for any discrepancies between our previous response and the submitted manuscript. We thoroughly verified the document and ensured that the current version aligns with the requested changes.
2. Reorganization of Section 3: As suggested, we subdivided Section 3 (“Biostimulants in Lettuce Cultivation”) into five thematic subparts (3.1 to 3.5), improving structure and reader navigation.
3. Title correction: The section formerly titled “2. Lactuca sativa L.” was renamed to “2. Biostimulants in Lettuce Cultivation” and was edited to focus on the scope of the review, eliminating excessive generalities on plant biology and industrial development.
4. Redundancy reduction: Repetitive content regarding lettuce cultivation challenges and importance was reviewed and condensed to avoid unnecessary duplication.
5. Table 1 expansion and integration: Table 1 was expanded to include more recent studies and better reflect the diversity of biostimulants. Additionally, we reduced the redundancy between the table and the main text by summarizing textual descriptions more succinctly.
6. Analytical depth and synthesis: The revised text presents more critical synthesis of the reviewed literature, including discussion of observed trends, research gaps, and future perspectives.
7. Suggestion of modes of action: We added a paragraph summarizing the general mechanisms of action of biostimulants and proposed this as a potential figure for future inclusion.
8. Additional improvements: We also reformatted the references to comply with the journal’s style, included new studies published between 2020 and 2024, and removed generic mentions of productivity and production challenges, focusing instead on more analytical contributions.
We deeply appreciate your time and the depth of your feedback, which played a pivotal role in enhancing the scientific contribution of our work.
Round 2
Reviewer 3 Report (Previous Reviewer 4)
Comments and Suggestions for Authors
The authors have made substantial revisions to the manuscript, which has now essentially reached publishable standards. It would be even more ideal if a schematic diagram summarizing the comprehensive mechanisms of action could be added to complement the textual review.
Author Response
Dear Reviewer,
First, I'd like to thank you for your feedback on the resubmitted version. Based on the review comments, we created some diagrams. I'd like to know which of the options in the attached file are best suited to the article. If you have a diagram template you'd like to suggest, I'd be grateful.
Thank you very much.

Round 3
Reviewer 3 Report (Previous Reviewer 4)
Comments and Suggestions for Authors
The authors have made good revisions to the manuscript, and the current version has basically reached the level for publication. It would be even more ideal if the Conclusion section could be more specific rather than too general.
Author Response
Dear reviewer,
Thank you for your feedback. We've made the requested corrections.
This manuscript is a resubmission of an earlier submission. The following is a list of the peer review reports and author responses from that submission.
Round 1
Reviewer 1 Report
Comments and Suggestions for Authors
In the paper "Application Potential of Biostimulants in Hydroponic Lettuce", the author reviews the positive effects of biostimulants on various physiological processes and metabolism of hydroponic lettuce. For example, nitrogen-containing compound biostimulants are related to plant hormone synthesis, antioxidation, and biomass accumulation. This review will provide valuable information for the hydroponic cultivation of lettuce. However, it needs to be further improved before it could be published.
1. While the main text comprehensively covers aspects such as the botanical classification, morphology, and economic value of lettuce, it is important to note that these details are absent in the abstract. It is advisable to include relevant information about the lettuce's characteristics mentioned in the text, at least in a summarized form, in the abstract to offer readers a quick and accurate understanding of the overall research content related to lettuce. This will enhance the clarity and effectiveness of the abstract in presenting the essence of your work.
2. In the line 135-148, I noticed that the two passages you provided seem to have some degree of redundancy. In both paragraphs, you repeat the description of the characteristics and advantages of the hydroponic system in Brazil compared to soil-based systems. It would be beneficial to streamline the text to avoid this repetition and make it more concise and efficient. You could consider merging or rephrasing the relevant information to present a more coherent and less repetitive account of the hydroponic system's features and its superiority over traditional soil-based cultivation methods. This will enhance the readability and clarity of your work.
3. I noticed that the reference format in your manuscript is inconsistent and some information may be missing. Please ensure all references are in the correct format and complete with necessary details according to the required style. This is important for the quality of your work.
Reviewer 2 Report
Comments and Suggestions for Authors
This paper try to review the use of biostimulants in Lettuce production in a Hydroponic system.Most hydyoponic cultivation maybe used in plastic tunnel or greenhouse, so the climate extemes maybe decrease by facility, I think it is important to list the biostimulants treatment in tables to compare their different effects on Lettuce after treatments, so the part 5 should be divided into more parts to show the different effects of different biostimulants on plants growth and its physiological defense. The treatment methods also need to write out, so I suggest the paper should be revised or rewrite, so that it maybe accepted.
Reviewer 3 Report
Comments and Suggestions for Authors
Sentence 12,13: Rewrite this sentence: Because of its composition, which contains a substance with antioxidant action and beneficial effects on health, its consumption is constant. because now it makes no sense.
Key words: Provide 5 key words.
Line 31: Is lettuce really that nutritious? It has very few vitamins and minerals compared to other vegetables.
Lines 32-34: This statement is not entirely true. Lettuce is very low in phenolics.
Line 36: Remove the decimals, they are useless.
Line 54: This is a review article, not an original scientific article: Remove the word study.
Line 57: Remove the space.
Line 110: Remove all decimals.
Line 128: Again, it contains all the biochemicals, but in such small amounts that it is hard to say there are any health benefits.
Line 128: All references must be in MDPI style. Now the text is incorrect.
Line 139: Remove spaces.
Lines 181 to 189: Why is this needed in this review article? This is too basic and not needed. I suggest removing it entirely.
5. Biostimulants in hydroponic lettuce production
There are several major concerns here:
- The references cited are 10 years or older, which is not good for a review.
- The authors provide only some general data about biostimulants, but very little is discussed about the effects of biostimulants on lettuce.
-Only algal extracts are described in lettuce: So the title is not good. Then it should be stated: Potential of using algal extract biostimulants in lettuce production in a hydroponic system.
Conclusion: The conclusions are bad, with only basic description.
Reviewer 4 Report
Comments and Suggestions for Authors
The manuscript summarized the botanical characteristics, cultivation status, hydroponic system, and application of biosimulants of lettuce, and the results were of certain value. But, most of the manuscript was a brief summary of references, lacking the author's independent thinking and summarization. Some of the content was not related to the title and fail to meet the basic requirements of a review. My main suggestions and comments are as follows.
1. The abstract needed to clearly list the core conclusions of the paper, but the current version does not reflect any relevant information.
2. The hydroponic technology for lettuce was very mature and has been widely commercialized. The author should explain as much as possible what the value or novelty of this paper was.
3. There was almost no correlation between the first two parts of the review (Botanical Classification and Pathology, Economic Importance) and the title. It is recommended to delete them.
4. In fact, I believed that the correlation between the fourth part (4. Hydroponic Growth System and Nutrient Solution) and the title was also very low. Can be deleted.
5. As a review, all content should be closely centered around the title, and nearly 50% of the content in this manuscript had no clear correlation with the title.
6. Even for a review paper, it was not appropriate to directly cite large paragraphs of references. As shown on pages 4 and 5, almost all of them were directly quoted.
7. Only Part 5 (5. Biostimulants in Hydroponic Lettuce Cultivation) was the main content of the paper and needed to be written under different subheadings. The current version had poor logic and organization.
8. The conclusion of the manuscript was more like an introduction to a review, without presenting any substantive conclusions.
9. In short, the manuscript fail to meet the basic requirements of a review paper, lacking the author's own perspectives and analysis, and most of the content was only a simple summary of references.